# Study of Precipitates in Oxide Dispersion-Strengthened Steels by SANS, TEM, and APT

Sergey V. Rogozhkin [1,2,*], Artem V. Klauz [1,2], Yubin Ke [3], László Almásy [4], Alexander A. Nikitin [1,2], Artem A. Khomich [1,2], Aleksei A. Bogachev [1,2], Yulia E. Gorshkova [5,6], Gizo D. Bokuchava [5], Gennadiy P. Kopitsa [7] and Liying Sun [8]

1 Institute of Nuclear Physics and Engineering, National Research Nuclear University "MEPhI", 115409 Moscow, Russia; kav052@campus.mephi.ru (A.V.K.); nikitin@itep.ru (A.A.N.); artem.khomich@gmail.com (A.A.K.); bogachev@itep.ru (A.A.B.)
2 National Research Center "Kurchatov Institute", 123182 Moscow, Russia
3 Spallation Neutron Source Science Center, Dongguan 523803, China; keyb@ihep.ac.cn
4 Institute for Energy Security and Environmental Safety, HUN-REN Centre for Energy Research, 1121 Budapest, Hungary
5 Joint Institute for Nuclear Research, 141980 Dubna, Russia; yulia.gorshkova@jinr.ru (Y.E.G.); gizo@nf.jinr.ru (G.D.B.)
6 Institute of Physics, Kazan Federal University, 420008 Kazan, Russia
7 Petersburg Nuclear Physics Institute Named by B.P. Konstantinov of NRC "Kurchatov Institute", 188300 Gatchina, Russia; kopitsa@lns.pnpi.spb.ru
8 Institute of New Materials, Guangdong Academy of Sciences, Guangzhou 510650, China; sunliying@gdinm.com
* Correspondence: svrogozhkin@mephi.ru or sergey.rogozhkin@itep.ru

**Abstract:** In this work, the nanostructure of oxide dispersion-strengthened steels was studied by small-angle neutron scattering (SANS), transmission electron microscopy (TEM), and atom probe tomography (APT). The steels under study have different alloying systems differing in their contents of Cr, V, Ti, Al, and Zr. The methods of local analysis of TEM and APT revealed a significant number of nanosized oxide particles and clusters. Their sizes, number densities, and compositions were determined. A calculation of hardness from SANS data collected without an external magnetic field, or under a 1.1 T field, showed good agreement with the microhardness of the materials. The importance of taking into account two types of inclusions (oxides and clusters) and both nuclear and magnetic scattering was shown by the analysis of the scattering data.

**Keywords:** oxide dispersion-strengthened steel; ODS steel; small angle neutron scattering; SANS; transmission electron microscopy; TEM; atom probe tomography; APT; oxide particle; cluster

## 1. Introduction

Oxide dispersion-strengthened (ODS) alloys are the most well-known nanostructured structural materials. ODS steels are reinforced by a significant number of uniformly distributed nano-oxides and have considerably higher heat resistance than conventional alloys [1]. That is why ODS alloys are widely used for heat-resisting applications (e.g., turbine components [2], etc.). Stable oxide inclusions dispersed in metallic matrices improve both creep resistance at high temperatures and radiation resistance [3–5]. Therefore, these materials are also developed for nuclear engineering, for example, structural materials of advanced nuclear power plants as fast neutron reactors, future fusion reactors, and for other Generation IV reactors [6–13].

Enhanced properties of ODS alloys greatly depend on their nanostructure characteristics: the chemical composition, size, and spatial distribution of disperse inclusions. The nanostructure of ODS alloys consists of stoichiometric oxide particles detected by TEM [14–19] and nanoclusters detected by APT (see, e.g., articles [20–22] and reviews [23–26].

It is generally accepted that oxides increase the heat resistance and radiation resistance of steels, but the role of clusters is not so obvious. Moreover, in nano-oxide-strengthened steels, it is almost impossible to separate nano-oxides from clusters, and it is assumed that, in these steels, TEM and APT can detect the same objects [25,27]. An important role of nanoclusters was shown in [28,29] in irradiated ODS steels, where irradiation led to the growth of clusters and transformed them into fine oxides. Thus, nanoclusters can affect the stability of the nanostructure of ODS steels.

Clusters and oxides can be detected simultaneously by small-angle X-ray [30,31] and neutron scattering [32–36] (SAXS and SANS, respectively). Moreover, SAXS and SANS can study substantially larger volumes of material in comparison with TEM and APT. Therefore, these scattering methods provide more correct average characteristics, such as average size or number density of inclusions. It is assumed that the composition of nanoclusters in ODS steels differs from that of stoichiometric oxides and the clusters can contain more than 50% Fe and Cr. This difference in the contrast between the matrix and precipitates strongly affects the scattering intensities. Due to the magnetic properties of nanosized clusters (since they contain large amount of Fe and Cr), they can be separated from oxide particles, which are known to be non-magnetic, by applying a magnetic field during the SANS measurement. Moreover, ODS ferritic/martensitic steels are ferromagnetic materials, and the saturating magnetic field used in a SANS experiment makes it possible to correctly take into account the magnetic properties of these steels [17–20].

The steels chosen for the present study were Eurofer ODS, 13.5Cr-Fe$_3$Y ODS, and KP-4 ODS. These steels have different allowing systems (Cr, V, Al and Zr content) and, therefore, different proportions of oxides, cluster densities, and size distributions of nanoinclusions. A combination of different microscopic techniques (TEM and APT) allowed us to obtain the characteristics of inclusions (scatterers of neutrons) in the ODS steels for the analysis of SANS data. Furthermore, we evaluated the contributions of clusters and oxides to the hardness in comparison with the experimentally measured case.

## 2. Materials and Methods

The study investigated three oxide dispersion-strengthened (ODS) ferritic/martensitic steels: Eurofer ODS, 13.5Cr-Fe$_3$Y ODS, and KP-4 ODS (ODS 9Cr-V-W-Mn, ODS 13.5Cr-W, and ODS 15Cr-W-Al-Zr, respectively). The Eurofer ODS is a type of European steel produced by Plansee (the "EU batch"); the 13.5Cr-Fe$_3$Y ODS steel was developed at the Karlsruhe Institute of Technology; and the KP-4 ODS steel was developed in Japan by Kyoto University. These ODS steels were produced by mechanical alloying. Table 1 shows the composition of the studied materials.

**Table 1.** Chemical compositions of studied ODS steels, at. % (balance in Fe).

| Steel | Al | Ni | Zr | Mn | Cr | W | Y | O | V | C | N | Si |
|---|---|---|---|---|---|---|---|---|---|---|---|---|
| Eurofer ODS | - | 0.02 | - | 0.39 | 9.8 | 0.34 | 0.13 | 0.34 | 0.22 | 0.40 | 0.21 | 0.06 |
| KP-4 ODS | 7.56 | - | 0.19 | - | 15.9 | 0.58 | 0.16 | 0.57 | - | - | - | - |
| 13.5Cr-Fe$_3$Y ODS | - | - | - | - | 14.6 | 0.6 | 0.3 | | - | - | - | - |

It is important to note that the studied ODS steels were ferritic martensitic steels and ferromagnetic materials. The production differences in the studied steels are shown in Table 2.

Specimens of the ODS steels were studied via small-angle neutron scattering (SANS) diffractometers with unpolarized neutrons at the BNC reactor ("Yellow Submarine" diffractometer, YS-SANS, https://www.bnc.hu/ys-sans (accessed on 8 December 2023)) [37] and at the China Spallation Neutron Source (CSNS-SANS diffractometer, http://english.ihep.cas.cn/csns/ (accessed on 8 December 2023)) [38,39]. For SANS analysis, $17 \times 17 \times 2$ mm$^3$ samples were prepared from the initial materials. The YS-SANS experiment was carried out without a magnetic field. Measurements were carried out on this diffractometer at two

neutron wavelengths, λ = 0.42 and 1.01 nm, Δλ/λ = 18%, with two sample–detector distances, 1.3 and 5.6 m. The measurements at CSNS-SANS were performed under an external magnetic field (1.1 T and 5 mT), perpendicular to the incident neutron beam. The sample size was $7 \times 7$ mm$^2$ (Figure 1). Notably, the 5 mT magnetic field had basically no effect on the measurement and should be considered as having no field present. For CSNS-SANS, neutrons with wavelength ranges 1~10 Å were used, with a sample-to-detector distance of 4 m. The geometry used for the experiments provided accessible scattering vector ranges: $0.06 < Q < 4.5$ nm$^{-1}$ at YS-SANS and $0.05 < Q < 7.0$ nm$^{-1}$ at CSNS-SANS.

**Table 2.** Production details of studied ODS steels.

| Steel | Alloying Type | Thermomechanical Treatment |
|---|---|---|
| Eurofer ODS | Mechanical alloying of metal and $Y_2O_3$ powders | Normalized at 1100 °C for 30 min with water quenching, followed by tempering at 750 °C for 2 h with air cooling |
| KP-4 ODS | Mechanical alloying of metal and $Y_2O_3$ powders | Encapsulated in a soft steel capsule and degassed in vacuum at 400 °C for 3 h |
| 13.5Cr-Fe$_3$Y ODS | Yttrium was introduced through the Fe$_3$Y intermetallic powder, and oxygen was introduced through the oxidized powder of the matrix steel | Degassed for 4 h at 400 °C, then subjected to hot isostatic pressing at 1150 °C for 2 h at 100 MPa |

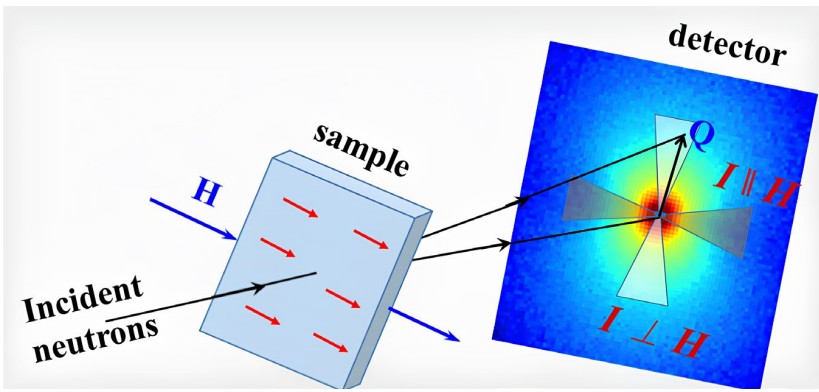

**Figure 1.** Schematic of the SANS experiment setup.

The raw data were corrected for background scattering, detector efficiency, and sample transmission. The measured scattering intensity was set to absolute scale by normalizing to the incident flux.

Analysis of the phase composition of ODS steels was carried out using TEM, electron diffraction, and scanning transmission electron microscopy. A Titan 80–300 S/TEM microscope (Thermo Fisher Scientific, Waltham, MA, USA) with an accelerating voltage of 300 kV was utilized. A ring high-angle dark-field detector (HAADF, Fischione) was used to obtain microphotographs in Z-contrast mode. The characteristic grain sizes and sizes of inclusions of different types were determined. More than 2000 detected objects were collected to obtain inclusion size distributions. The acquired size distributions of the inclusions allowed for the calculation of their average sizes, number densities, and respective standard deviations. To calculate the uncertainty of these values, the studied layer thickness, the resolution of the microscope, and the value deviation were taken into account.

Samples for the TEM study were prepared with a focused ion beam (FIB) with Ga+ ions using a dual-beam scanning electron microscope, HELIOS NanoLab 600 (FEI, Eindhoven, Holland), at an accelerating voltage of 5–30 kV. Considering that this process created a damaged layer due to interaction with the ion beam, finalizing thinning was performed at an accelerating voltage of 2 kV. Thin cross-section samples were prepared for TEM studies in this way.

Local chemical composition ODS steels at the nanometer scale were investigated by the femtosecond laser evaporation tomographic atom probe APPLE-3D, created at the Institute of Theoretical and Experimental Physics (Moscow, Russia) [39]. Data were collected at a reference sample temperature of 40–50 K in the laser evaporation mode with a wavelength of 515 nm, a laser pulse duration of 300 fs, a frequency of 25 kHz, and a pulse energy of 0.1–1.2 µJ [40]. The pressure in the research chamber was $(5 \div 7) \times 10^{-10}$ Torr.

Atom probe tomography samples were firstly prepared with volumes of $300 \times 300 \times 10,000$ µm$^3$ by means of electroerosion cutting in water. The next step was carried out through standard methods of electrochemical anodic electropolishing to form the tips of the samples, with rounding radii of 15–50 nm. The obtained needle-samples were checked with a JEOL 1200 EX transmission electron microscope before the APT study.

For each material, no fewer than two $30 \times 30 \times 300$ nm$^3$ volumes were acquired during the APT study. APT data analysis included mass spectrum interpretation and characterization of three-dimensional distributions of chemical elements in the studied volumes. The general Bass reconstruction method was used to reconstruct 3-D atom maps, in which the back projection of each detected ion was calculated using the radius of the sample tip and the distance between the sample and the detector [41].

## 3. Results

### 3.1. APT and TEM Results

The investigated steels consisted of ferritic grains with sizes ranging from 300 nm to 2 µm in Eurofer ODS steel, from 300 nm to 3 µm in KP-4 ODS steel, and from 250 nm to 2 µm in 13.5Cr-Fe$_3$Y ODS steel.

Small (5–6 nm) oxide particles were detected in the studied steels using transmission electron microscopy (TEM). A small number of larger (44–55 nm) oxide particles were also present, and in Eurofer ODS steel, large carbides of the M$_{23}$C$_6$ type were also detected. Chemical analysis of Eurofer ODS was conducted in an earlier work [27], while new bright-field images are presented in Figure 2. Images of oxide particles in KP-4 ODS and 13.5Cr-Fe$_3$Y ODS steels, acquired using high-angle annular dark-field (HAADF) mode, are presented, as well as their chemical mapping, in Figures 3 and 4. To analyze the number density of the inclusions, the local thicknesses of the TEM specimens were measured by electron energy loss spectroscopy (EELS) using the zero-loss peak. For each sample, at least six thickness measurements were performed at different positions. For Eurofer ODS, the average value was ~60 nm, while in KP-4 ODS and 13.5Cr-Fe$_3$Y ODS, it was ~65 nm. The average size of the small oxide particles was $6 \pm 2$ nm in Eurofer ODS, $5 \pm 2$ nm in KP-4 ODS, and $6 \pm 1$ nm in 13.5Cr-Fe$_3$Y ODS, with number densities of $4 \pm 1 \times 10^{22}$, $2 \pm 1 \times 10^{22}$, and $0.8 \pm 0.2 \times 10^{22}$ m$^{-3}$ for Eurofer ODS, KP-4 ODS, and 13.5Cr-Fe$_3$Y ODS, respectively. Figure 5 presents size distributions for these oxide particles.

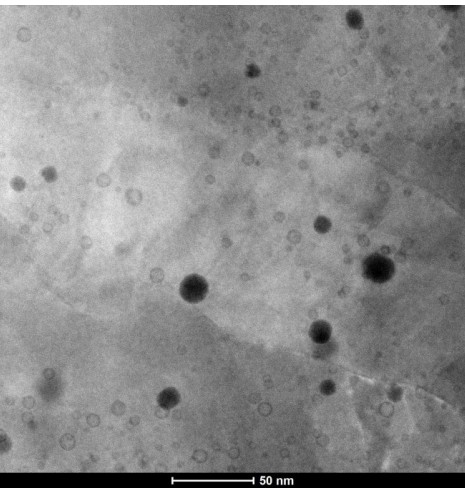

**Figure 2.** Bright-field TEM image of Eurofer ODS.

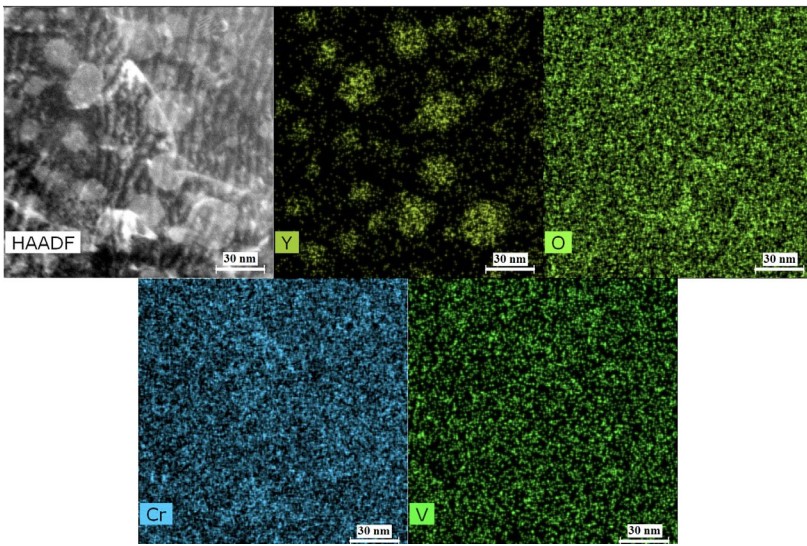

**Figure 3.** Chemical mapping of 13.5Cr-Fe₃Y ODS.

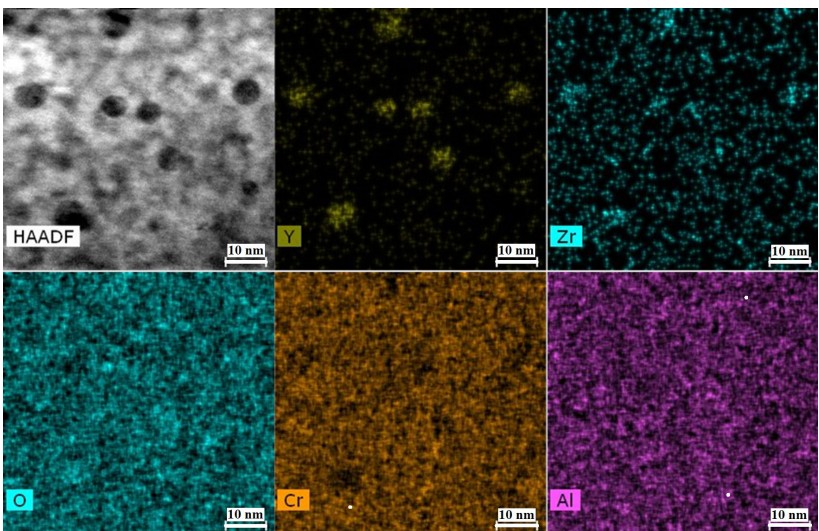

**Figure 4.** Chemical mapping of KP-4 ODS.

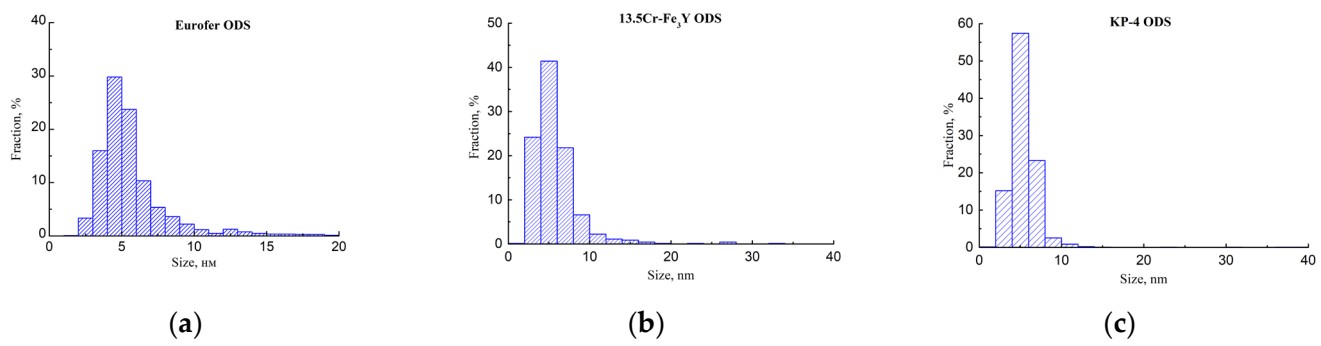

**Figure 5.** Oxide particle size distributions in: (**a**) Eurofer ODS; (**b**) 13.5Cr-Fe₃Y ODS; (**c**) KP-4 ODS.

Based on chemical analysis, using EDX mapping, the most probable types of oxides were identified with the consideration of ODS steels in previous works. Eurofer ODS is the European steel for fusion program. TEM analysis of large oxides was carried out carefully in multiple works [42,43], and 13.5 Cr ODS alloys were designed in Karlsruhe Institute of Technology to increase the corrosion resistance ODS steels for fusion [44–46].

ODS steels based on FeCrAl matrix metal were widely designed in Japan [47,48]. Overall, the stoichiometry of the oxides in Eurofer ODS and 13.5Cr-Fe$_3$Y ODS steels was Y$_2$O$_3$, while in KP-4 ODS steel, it was Y$_4$Zr$_3$O$_{12}$.

Atom probe tomography revealed the high number density of the nanoclusters. Figures 6–8 show atom maps of the studied materials. The average cluster size varied from 2 to 3 nm, and the number density ranged from $1 \times 10^{23}$ m$^{-3}$ to $3 \times 10^{23}$ m$^{-3}$. Cluster size distributions are shown in Figure 9. The values for number density and average size are given in Table 3.

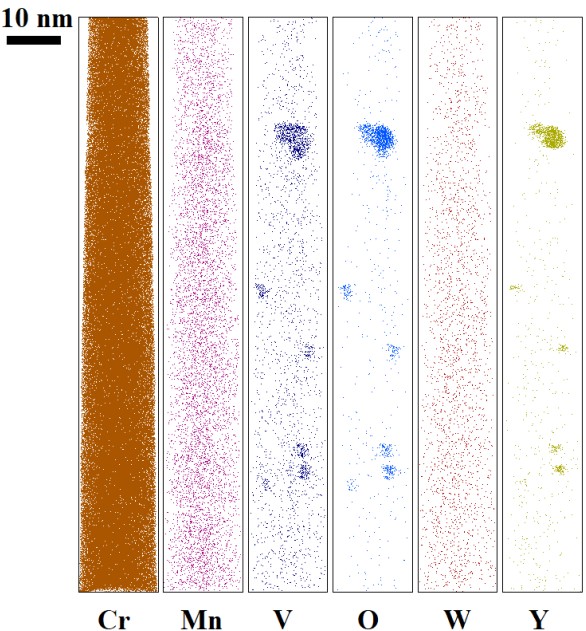

**Figure 6.** Atom maps of Eurofer ODS.

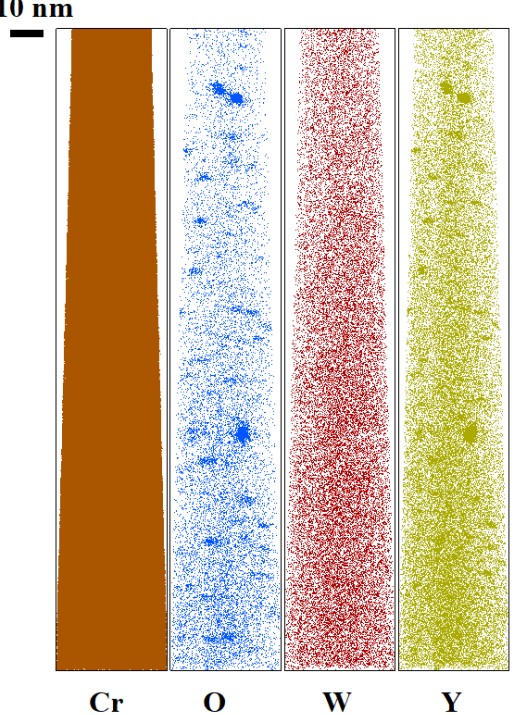

**Figure 7.** Atom maps of 13.5Cr-Fe$_3$Y ODS.

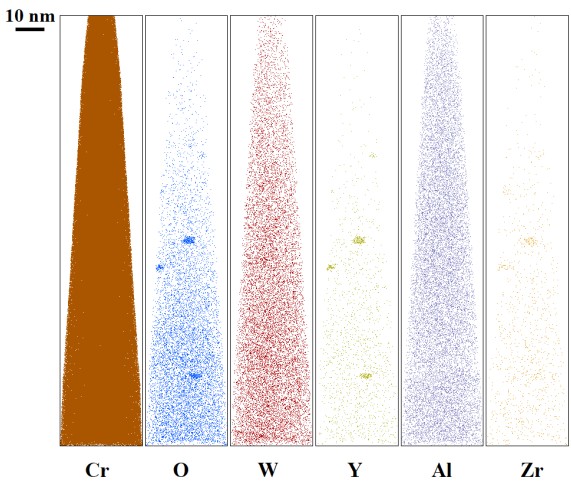

**Figure 8.** Atom maps of KP-4 ODS.

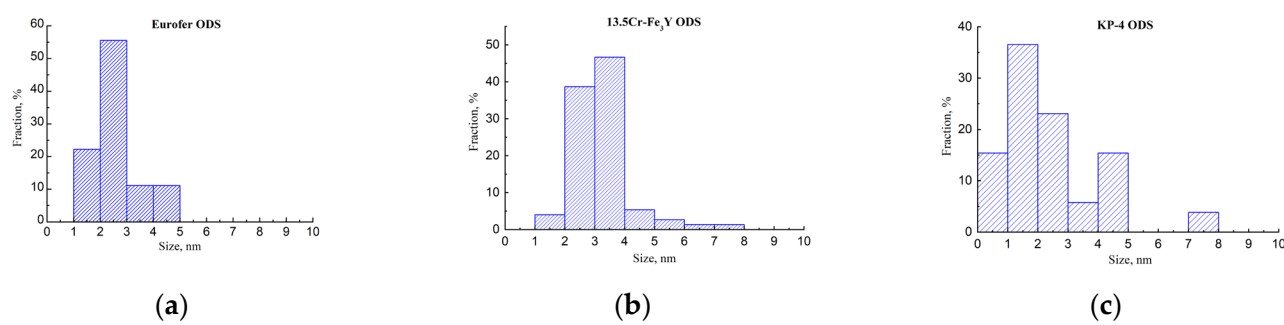

| (a) | (b) | (c) |

**Figure 9.** Cluster size distributions in: (**a**) Eurofer ODS; (**b**) 13.5Cr-Fe$_3$Y ODS; (**c**) KP-4 ODS.

**Table 3.** Relevant TEM and APT results for studied materials.

| Material | Phase | Chemical Composition/Formula | Average Size | Number Density, $10^{22}$ m$^{-3}$ | $\Delta\rho^2$ (SLD) Nuclear, $10^{21}$ | $\Delta\rho^2$ (SLD) Magnetic, $10^{21}$ | A |
|---|---|---|---|---|---|---|---|
| | Matrix | Fe89 Cr10 Mn0.4 V0.2 N0.1 O0.1 W0.3 Y0.1 | - | - | - | - | - |
| Eurofer ODS | Grains Oxides | - Y$_2$O$_3$ | 0.5 ± 0.2 μm 6 ± 2 nm | - 4 ± 1 | - 0.98 | - 2.3 | - 3.3 |
| | Clusters | Fe46 Cr20 V9 O11 Y10 W0.3 Mn0.5 N0.1 | 2 ± 1 nm | 32 ± 5 | 0.43 | 0.24 | 1.6 |
| | Matrix | Fe85 Cr13 Mn0.7 O0.2 V0.1 Y0.4 Ni0.2 Si0.1 W0.4 | - | - | - | - | - |
| 13.5Cr-Fe$_3$Y ODS | Grains Oxides | - Y$_2$O$_3$ | 0.8 ± 0.4 μm 6 ± 1 nm | - 0.8 ± 0.2 | - 0.9 | - 2.2 | - 3.4 |
| | Clusters | Fe50 Cr14 Mn0.5 O6 Y11 Ni0.6 Si0.2 W0.11 | 2 ± 1 nm | 32 ± 4 | 0.5 | 0.27 | 1.5 |
| | Matrix | Fe76 Cr16 O0,4 Y0,1 Al7 Zr0,2 W0,6 | - | - | - | - | - |
| KP-4 ODS | Grains Oxides | - Y$_4$Zr$_3$O$_{12}$ | 0.6 ± 0.4 μm 5 ± 2 nm | - 2 ± 1 | - 0.9 | - 2.2 | - 3.4 |
| | Clusters | Fe49 Cr12 O19 Y12 Al5 Zr3 | 3 ± 1 nm | 10 ± 2 | 0.5 | 0.27 | 1.5 |

Figure 10 demonstrates cluster enrichments ordered by their numbers from the largest to smallest (number one being the largest, etc. Where there were more than 50 clusters, only every 2nd or 3rd cluster is shown). Element concentrations in the matrix, as well as

cluster average values (determined with APT), are shown in Table 3. Clusters in KP-4 ODS and 13.5Cr-Fe$_3$Y ODS were significantly enriched in Y and O, while KP-4 ODS (which had a high concertation of Al in the initial state) had an Al deficiency in the clusters.

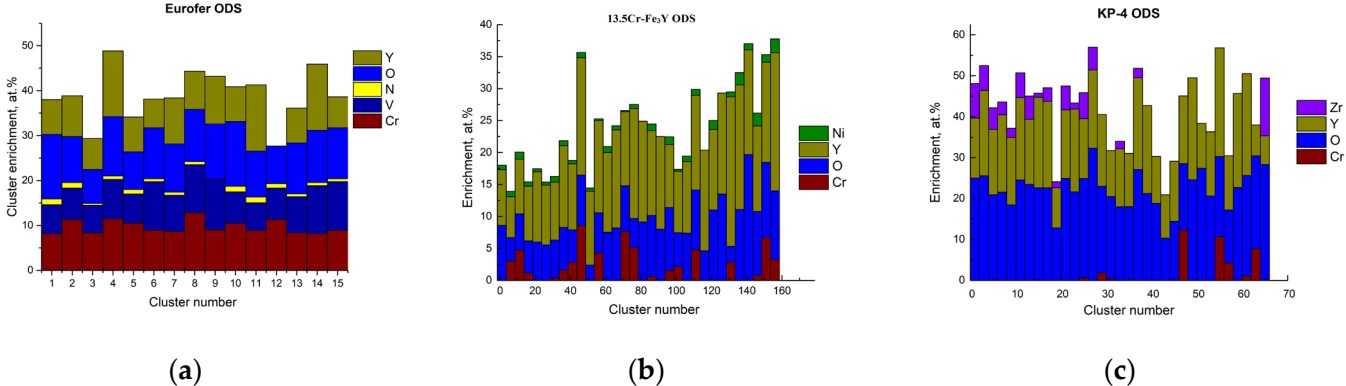

**Figure 10.** Enrichment of clusters in the ODS steels: (**a**) Eurofer ODS; (**b**) 13.5Cr-Fe$_3$Y ODS; (**c**) KP-4 ODS.

It must be also noted that, in 13.5Cr-Fe$_3$Y ODS, APT showed the presence of a number of elements (Mn, V, Al, Ni, Si, and P) which were not declared in the initial material specification.

### 3.2. SANS Results

An example of SANS intensities on a 2D detector is shown in Figure 11. The scattered intensity without a magnetic field was practically direction-independent. Under the applied field, the scattered intensity was anisotropic due to the magnetic scattering. This allowed for the separation of the magnetic and non-magnetic contributions.

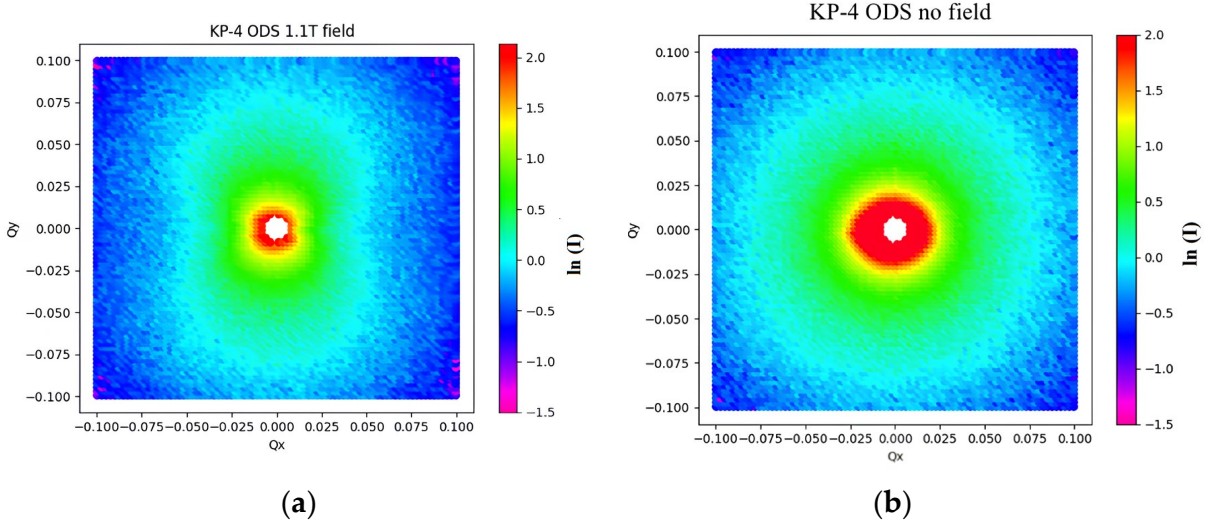

**Figure 11.** SANS intensity, measured with CSNS-SANS on the KP-4 ODS steels: (**a**) under 1.1 T magnetic field applied horizontally; (**b**) without magnetic field.

A comparison of radially averaged intensities with and without a magnetic field is shown in Figures 12 and 13 for CSNS-SANS and YS-SANS, respectively. In the high Q range beyond 0.02 Å$^{-1}$, all the curves were nearly field-independent. Therefore, the hump appearing at the 0.09 Å$^{-1}$ position indicates the chemical heterogeneity, with a size of around 7 nm. In comparison, the second hump at 0.008 Å$^{-1}$, observed in samples without magnetic fields, disappeared under 1.1 T, indicating its magnetic nature. It was inferred that the strong field-dependent scattering features may have resulted from the magnetic domains with sizes of ~80–100 nm, which can be eliminated by applying a field like 1.1 T in

this study. The iron–chromium matrix of the studied steels was ferromagnetic, and it is possible that the domains mentioned above were magnetic domains of the ferritic/martensitic steels or magnetically ordered regions near grain boundaries or large inclusions.

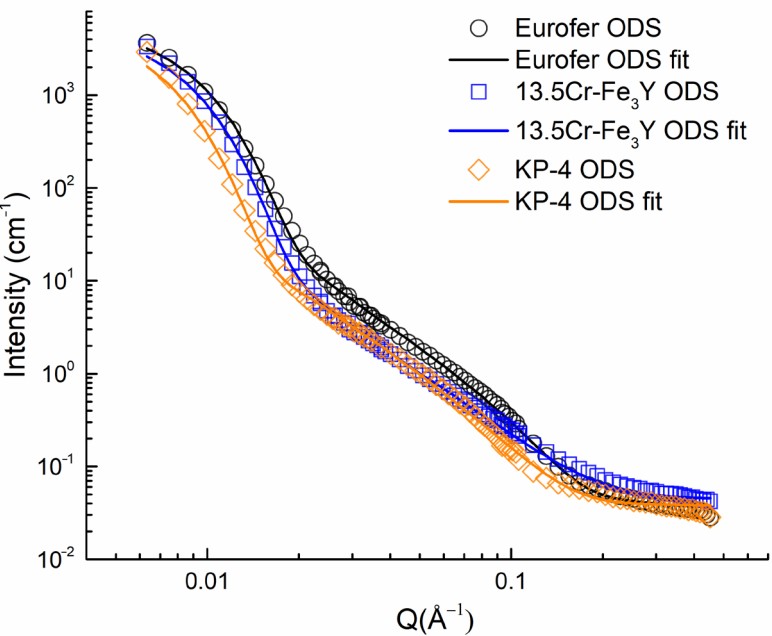

**Figure 12.** Small–angle neutron scattering obtained at YS-SANS for Eurofer ODS, 13.5Cr-Fe$_3$Y ODS, and KP-4 ODS.

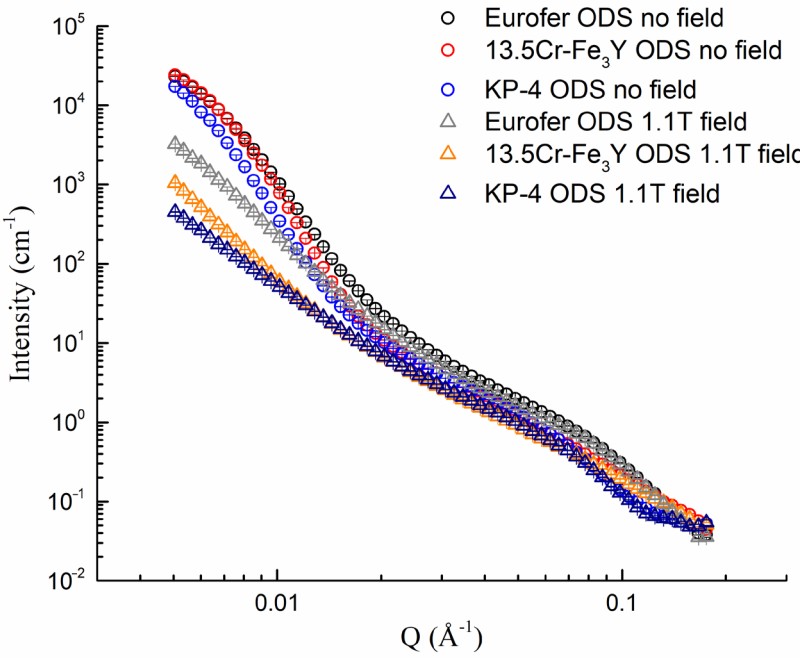

**Figure 13.** Small–angle neutron scattering obtained at CSNS-SANS for Eurofer ODS, 13.5Cr-Fe$_3$Y ODS and KP-4 ODS.

Under a 1.1 T magnetic field, the intensities for the two directions were integrated within a 20 degree sector in each direction ($\perp$ and $\parallel$ to H). The resulting intensities are shown in Figure 14.

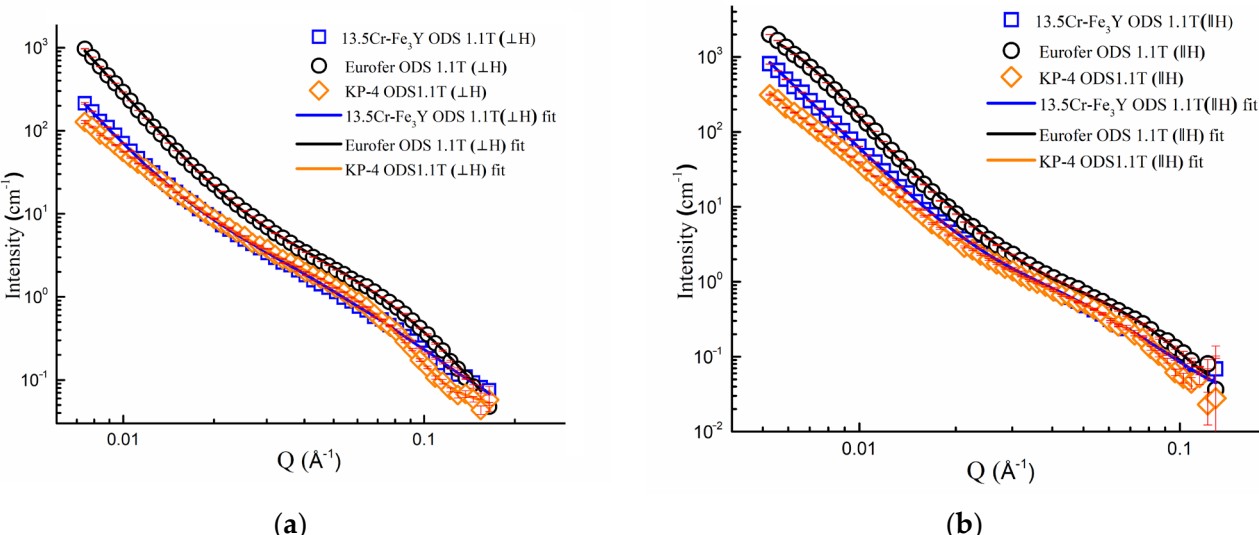

**Figure 14.** Small-angle neutron scattering intensities obtained at CSNS-SANS for Eurofer ODS, 13.5Cr-Fe$_3$Y ODS, and KP-4 ODS under 1.1 T field in two directions (within a 20-degree angle) as well as their fits: (**a**) parallel to the magnetic field; (**b**) perpendicular to the magnetic field.

To quantitatively analyze the nanostructures of the three steels, the SANS data were fitted using the combined models provided in the Irena SANS package for the Igor Pro 9.0 software [49].

In the calculation model, two types of particles were assumed, and the contrast values, $\Delta\rho_{ox}$ for oxides and $\Delta\rho_{cl}$ for clusters, were calculated. The scattering intensity in function of the momentum transfer was approximated by the following formula:

$$I(Q) = C\,exp\left(-\frac{Q^2 R^2_g}{3}\right) + \left((\Delta\rho^{ox}_{nucl})^2 +\right.$$
$$(\Delta\rho^{ox}_{mag})^2 sin^2\alpha)\int_0^\infty n_{ox}(R)V_{ox}(R)^2 F(Q,R)^2 dR + ((\Delta\rho^{cl}_{nucl})^2 + \tag{1}$$
$$\left.(\Delta\rho^{cl}_{mag})^2 sin^2\alpha)\int_0^\infty n_{cl}(R)V_{cl}(R)^2 F(Q,R)^2 dR + B(1)\right.$$

Here, $Q$ is a scattering vector, defined as $Q = (4\pi\,sin\,\theta)/\lambda$, where $\lambda$ is the wavelength and $2\theta$ is the scattering angle. $B$ is the incoherent background. The first term is the Guinier scattering at small $Q$ from large, coarse precipitates [49,50]. The second term is the intensity scattered by a distribution of precipitates [36,51,52]. $\Delta\rho_{nucl,\,mag}$ is the magnetic (*mag*) and nuclear (*nucl*) scattering length density (SLD) difference (contrast) between the matrix and the precipitate; $\alpha$ is the angle with respect to the magnetic field direction. R is the radius of a spherical precipitate, $V(R) = \frac{4}{3}\pi R^3$ is the volume of an individual precipitate, $n(R) = \frac{N}{R\sigma\sqrt{2\pi}}exp\left(-\frac{(ln(R)-\mu)^2}{2\sigma^2}\right)$ is the lognormal size distribution of precipitates, $\int_0^\infty n(R)dR = N$ is the total number of precipitates, and $F(Q,R) = 3\frac{sin(QR)-QRcos(QR)}{(QR)^3}$ is their form factor. Equation (1) was used to calculate the size distributions of precipitates, their average sizes, and their number densities.

The results of previous TEM and APT analyses were used to constrain the size values for the oxide particles and the clusters, respectively. Equation (1) was used to analyze the SANS data obtained without and with a magnetic field. The scattering intensity in measurements without magnetic fields was fitted with contrast values $\Delta\rho^2 = \Delta\rho^2_{nucl} + 2/3\,\Delta\rho^2_{mag}$ [53,54], whereas for the measurements in 1.1 T, the two intensities ($\perp$ and $||$ to H) (Figure 13) were fitted with $\Delta\rho^2 = \Delta\rho^2_{nucl} + \Delta\rho^2_{mag}$ in the case of the direction $\perp$ to H and with $\Delta\rho^2 = \Delta\rho^2_{nucl}$ for the direction $||$ to H.

For an accurate analysis of the SANS data, complementary information on the chemical compositions of the clusters and the matrix was needed. An overview of the data obtained by TEM and APT and used for the SANS analyses is provided in Table 3. The $\Delta\rho_{nucl,mag}$ val-

ues were also calculated from these data using the method described in [36]. Table 3 shows also the A-factor values for different inclusions calculated with the following formula:

$$A(Q) = 1 + \frac{\Delta\rho_{mag}^2}{\Delta\rho_{nucl}^2} \qquad (2)$$

The fitting procedure was carried out through multiple iterations: while keeping the model's particle size close to the ones gathered from TEM and APT, a best possible fit was attempted with the modeling within the Irena SANS package for the Igor Pro software. If the values during the fit deviated too much from the ones seen by TEM and APT, another fitting iteration was attempted with different starting parameter values. To calculate the magnetic contribution, the 2D scattering patterns were integrated within a 20° angle along both the horizontal and vertical directions.

The results of SANS analysis are collected in Tables 4 and 5. Table 4 shows the results derived from the Guinier term of the model compared to the average sizes of large particles estimated from TEM images. The sizes of the larger oxides were estimated from a small number of objects on the TEM images and should be considered only as estimates. Despite some differences, the data of TEM and SANS were in a good agreement, considering that estimation of large oxide inclusions was difficult using TEM methods due to the low number density. In this respect, the analysis of the scattering curves in the region of small $Q$ allowed us to determine the average sizes of such inclusions with better accuracy.

**Table 4.** SANS and TEM results for large-sized particles.

| Steel | TEM Oxide Size, nm | $2R_g$ (No Field) $^{1/2*}$, nm | $2R_g$ (1.1 T Field) $^{2*}$, nm | |
|---|---|---|---|---|
| Eurofer ODS | ~40 | 46/35 | ($\parallel H$) | 39 |
| | | | ($\perp H$) | 37 |
| 13.5Cr-Fe₃Y ODS | ~50 | 50/40 | ($\parallel H$) | 45 |
| | | | ($\perp H$) | 43 |
| KP-4 ODS | ~50 | 54/46 | ($\parallel H$) | 46 |
| | | | ($\perp H$) | 43 |

* [1]—YS-SANS, [2]—CSNS-SANS.

**Table 5.** Comparison of SANS results obtained with different magnetic fields to TEM and APT.

| Steel | TEM | | APT | | SANS (No Field) $^{1/2*}$ | | | SANS (1.1 T Field) $^{2*}$ | | |
|---|---|---|---|---|---|---|---|---|---|---|
| | $d$, nm | $N$, $10^{22}$ m$^{-3}$ | $d$, nm | $N$, $10^{22}$ m$^{-3}$ | Object Type | $d$, nm | $N$, $10^{22}$ m$^{-3}$ | Azimuthal Angle | $d$, nm | $N$, $10^{22}$ m$^{-3}$ |
| Eurofer ODS | 6 ± 2 | 4 ± 1 | 2 ± 1 | 30 ± 5 | Oxide | 4/6 | 5/1.7 | ($\parallel H$) | 6 | 3.3 |
| | | | | | | | | ($\perp H$) | | 1.2 |
| | | | | | Cluster | 2/3 | 40/25 | ($\parallel H$) | 2 | 42 |
| | | | | | | | | ($\perp H$) | | 35 |
| 13.5Cr-Fe₃Y ODS | 6 ± 1 | 0.8 ± 0.2 | 2 ± 1 | 32 ± 4 | Oxide | 5/7 | 3.5/0.5 | ($\parallel H$) | 6 | 3.2 |
| | | | | | | | | ($\perp H$) | | 1.1 |
| | | | | | Cluster | 2/2 | 29/27 | ($\parallel H$) | 2 | 37 |
| | | | | | | | | ($\perp H$) | | 28 |
| KP-4 ODS | 5 ± 2 | 2 ± 1 | 3 ± 1 | 9 ± 2 | Oxide | 7/7 | 2/1.6 | ($\parallel H$) | 6 | 1.2 |
| | | | | | | | | ($\perp H$) | | 0.8 |
| | | | | | Cluster | 3/3 | 6/4.6 | ($\parallel H$) | 2 | 11.3 |
| | | | | | | | | ($\perp H$) | | 6.2 |

* [1]—YS-SANS, [2]—CSNS-SANS.

Comparison of various results of fitting the SANS data showed that the number densities of inclusions (both oxides and clusters) were usually higher when using data obtained in the direction of scattering along the applied magnetic field (when only nuclear contrast is taken into account). The use of scattering data in the direction perpendicular to the magnetic field, or for scattering without a magnetic field (when the magnetic domains in a ferromagnetic material are oriented randomly), allowed for slightly improvements to the result. Notably, since the stoichiometry of oxides was determined only by chemical analysis of large particles, it can be assumed that the chemical composition of small oxides (and their corresponding SLD) could differ from that of larger oxides. This point requires additional clarification and may lead to differences in the calculated number densities of the inclusions. In general, the data obtained by local TEM and APT methods agree fairly well with the results of the SANS analysis.

Table 5 presents the results of the fitting of SANS data compared to TEM and APT.

### 3.3. Analysis of Hardening

Data obtained from complex analysis of the morphology of the ODS steels allowed us to estimate the contributions of the different inclusions to the hardening. The dispersed barrier hardening model (DBH model) was used to estimate the yield strength (see, e.g., [55]). In this model, each barrier type contributes to hardening according to the Orowan formula:

$$\Delta\sigma_i = M_T \alpha_i \mu b \sqrt{N_i d_i} \tag{3}$$

where $\alpha_i$ is the barrier strength; $M_T$ is the Taylor factor; $\mu$ is the shear module; $b$ is the Burgers vector modulus; and $N_i$ and $d_i$ are the number density and average size of the barrier.

Barrier values $\alpha_i$ are different for oxide inclusions and clusters. According to data in the literature, the coefficients chosen for the calculations were $\alpha_c = 0.1$ for clusters [56] and $\alpha_o = 0.63$ for oxide inclusions [57].

In addition to small particles, grain boundaries and the matrix contributed to the total hardening. The strengthening from grain boundaries was determined by the well-known Hall–Petch relation:

$$\sigma_{gb} = kD^{-1/2}, \tag{4}$$

where $D$ is the grain size and $k$ = 338 MPa/μm.

Solid-phase hardening was considered for ferritic–martensitic steels: $\sigma_m$ = 0.255 GPa [58]. Total hardening from all barrier types was calculated using the following formula (see, e.g., [59]):

$$\sigma_y = \sqrt{\sigma_c^2 + \sigma_o^2} + \sigma_m + \sigma_{gb}, \tag{5}$$

where $\sigma_c$, $\sigma_o$, and $\sigma_{gb}$ represent the hardening from clusters, from oxide particles, and from grain boundaries, respectively, and $\sigma_m$ is the solid-phase hardening.

Hardening-to-hardness conversion was performed using the formula $H_v = 3 * \sigma_y$ [59]. The results of the hardness calculations are given in Table 6.

**Table 6.** Comparison of calculated hardness for TEM&APT and SANS data.

| Steel | TEM and APT, GPa | SANS (No Field) 1/2*, GPa | SANS (1.1 T, ∥H) 2*, GPa | SANS (1.1 T, ⊥H) 2*, GPa | Microindentation, $H_v$, GPa |
|---|---|---|---|---|---|
| Eurofer ODS | 4.2 ± 0.2 | 4.1 ± 0.2/3.6 ± 0.2 | 4.1 ± 0.2 | 3.3 ± 0.2 | 3.8 ± 0.2 |
| 13.5Cr-Fe₃Y ODS | 2.9 ± 0.2 | 3.6 ± 0.2/2.7 ± 0.3 | 3.7 ± 0.2 | 3.0 ± 0.3 | 3.0 ± 0.2 |
| KP-4 ODS | 3.3 ± 0.2 | 3.5 ± 0.2/3.4 ± 0.2 | 3.1 ± 0.3 | 2.9 ± 0.3 | 3.2 ± 0.3 |

* [1]—YS-SANS, [2]—CSNS-SANS.

### 4. Discussion

To identify the features of various material models in the analysis of SANS curves, the calculated hardness values were compared with the data on the micro-indentation of Eurofer ODS, 13.5Cr-Fe₃Y ODS, and KP-4 ODS steels (see Figure 15). The worst results

were demonstrated by models that took into account only one type of inclusion (oxides or clusters). The model which took into account only oxides and nuclear scattering gave significantly inflated values. Taking into account magnetic scattering somewhat improved the results. The model which took into account only clusters resulted in clearly underestimated hardness values. The closest values of hardness to the experimental data were demonstrated by a model with two types of inclusions for all considered scattering cases: along the external magnetic field (where there is a contribution of only nuclear scattering), perpendicular to the external magnetic field, and without an external field (where in addition to nuclear contrast there is a contribution of magnetic scattering). For all three steels, the best result for the calculated hardness was demonstrated by a model with two types of inclusions, using SANS data without an external magnetic field. One of the reasons for this may be that the applied field did not allow us to achieve the saturated magnetization of the steels. Note that, in a number of works, a significantly larger magnetic field (up to 1.5 T) was applied for the purpose of reaching saturation (e.g., [32]).

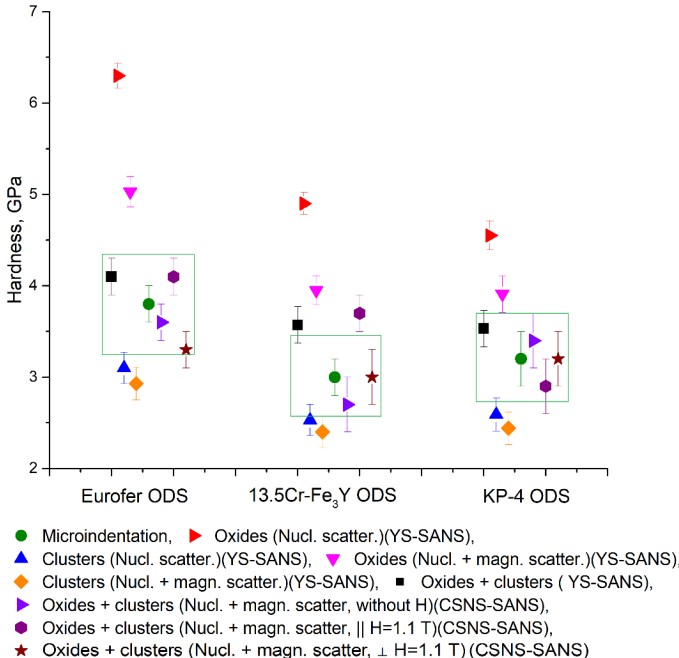

**Figure 15.** Calculated hardness values and data of micro-indentation of Eurofer ODS, 13.5Cr- $Fe_3Y$ ODS, and KP-4 ODS steels. Rectangles show areas of deviation by 15% from the measured microhardness.

## 5. Conclusions

Three ODS steels developed in Europe and Japan were studied by TEM and APT, as well as two SANS setups, YS-SANS and CSNS-SANS, using magnetic and non-magnetic small angle scattering. The number densities and sizes of the nanoscale oxide inclusions and clusters in these steels were analyzed. According to TEM and APT, the average size of oxide inclusions in these steels was about 5–6 nm, and the number density was between $0.8 \times 10^{22}$ m$^{-3}$ and $4 \times 10^{22}$ m$^{-3}$. The average size of the clusters in these steels was about 2–3 nm, and the number densities varied from $1 \times 10^{23}$ m$^{-3}$ to $3.2 \times 10^{23}$ m$^{-3}$. TEM and APT data allowed us to evaluate the chemical compositions of the oxides, clusters, and matrix. These data were used to calculate the nuclear and magnetic contrasts and the estimated size ranges of oxides and clusters, and were used further in the analysis of SANS data.

From the SANS experiments, multiple sets of data were obtained, either with no external magnetic field or with a field of 1.1 T. The number densities and the sizes of the particles in the studied materials were obtained by model fitting. The same scattering curves were fitted with an assumption that only one type of inclusion was present, as

well as that both clusters and oxides were present. The studied steels were ferromagnetic materials, and the role of magnetic scattering was evaluated in the analysis of SANS data.

The data obtained by different techniques and experiments were compared using the DBH model as well as microhardness measurements. It was shown that modeling the SANS data with both oxides and clusters was necessary to obtain a good agreement between the calculated hardness and the experimentally determined microhardness. It was also shown that the modeling of SANS measurement data, when performed without an external magnetic field or averaged in a sector perpendicular to the field direction, showed better agreement with the microhardness values, which suggests that both nuclear and magnetic contrasts of particles should be evaluated for the modeling of intensity curves.

Overall, the present study demonstrated that using SANS in a combination of TEM and APT leads to a better agreement between the hardness calculated based on the morphology obtained in the scattering experiments and the experimental hardness values.

**Supplementary Materials:** The following supporting information can be downloaded at: https://www.mdpi.com/article/10.3390/nano14020194/s1, Files S1–S3: YS-SANS raw data for 1D with no field; Files S4–S9: CSNS-SANS raw data for 1D with no field and 1.1 T field; Files S10–S15: CSNS-SANS raw data for 2D with no field and 1.1 T field.

**Author Contributions:** Conceptualization—S.V.R., Y.K. and L.A.; methodology—S.V.R. and A.V.K.; validation—A.A.N. and A.A.K.; formal analysis—S.V.R., A.V.K. and A.A.B.; investigation—Y.E.G., G.D.B., G.P.K., L.S. and A.V.K.; resources—S.V.R., Y.K. and L.A.; data curation—Y.E.G., L.S. and G.P.K.; writing—original draft preparation—A.V.K. and A.A.K.; writing—review and editing—S.V.R., Y.K. and L.A.; visualization—A.A.N. and A.A.B.; supervision—S.V.R. and G.D.B.; project administration—S.V.R.; funding acquisition—S.V.R. All authors have read and agreed to the published version of the manuscript.

**Funding:** This work was supported financially by the Ministry of Science and Higher Education of the Russian Federation (Agreement No. 075-15-2021-1352).

**Data Availability Statement:** The SANS 1D and 2D data that are presented in this study are available as Supplementary Materials. The APT and TEM data presented in this study are available upon request from the corresponding author. The raw data are not publicly available due to regulations of the NRC "Kurchatov Institute" on ongoing projects.

**Acknowledgments:** Tomographic atom-probe analysis was performed using the equipment of the KAMIKS Shared Use Center (http://kamiks.itep.ru (accessed on 8 December 2023)) of the NRC "Kurchatov Institute". Small-angle neutron scattering was performed using the equipment of the Budapest Neutron Centre and the China Spallation Neutron Source (Agreement No.: CSNS-US-2020-001). The authors thank P. Vladimirov from the Karlsruhe Institute of Technology (Germany) and A. Kimura from the Kyoto University (Japan) for providing samples of the ODS steels.

**Conflicts of Interest:** The authors declare no conflicts of interest.

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
