# Peer review of "Study of Precipitates in Oxide Dispersion-Strengthened Steels by SANS, TEM, and APT"

_nanomaterials, doi:10.3390/nano14020194_

Round 1

Reviewer 1 Report

Comments and Suggestions for Authors

The paper deals with ODS steels and is mainly focused on oxides and clusters. By means of APT, TEM and SANS, the authors have demonstrated that both oxides and clusters have to be taken into account to correctly reproduce the hardness property of the ODS alloys. 

The paper is weel written and organized and deserves publication. I have only some brief comments:

The scale in the figure 3 and 4 are too small to be correctly readen.

Line 276: APT instead of ART?

Line 151: How do you measure thickness to evaluate oxide density by TEM?

Line 156: can the authors show some EDS spectra to support oxide stoichiometry?

Figure 10: It is not clear what "cluster number" refer to?

In table 3, the clusters composition are determined by APT?

Author Response

We would like to thank the reviewers for their suggestions regarding the article. We have addressed the following issues:

The scale in the figure 3 and 4 are too small to be correctly readen.

Re: The scale has been increased for better viewing.

Line 276: APT instead of ART?

Re: This is indeed a typo and has been corrected

Line 151: How do you measure thickness to evaluate oxide density by TEM?

Re: This part has been added to the text to clarify the thickness measurement: “To analyze the number density of inclusions, the local thickness of the TEM specimens was measured by electron energy loss characteristic spectroscopy (EELS) using the zero-loss peak. For each sample, at least 6 thickness measurements were performed at different positions. For Eurofer ODS the average value was ~ 60 nm, in KP-4 ODS and 13.5Cr-Fe3Y ODS it was ~ 65 nm.”

Line 156: can the authors show some EDS spectra to support oxide stoichiometry?

Re: The key question of our TEM analysis was size distribution and number density of oxides inclusion. Because the ODS steels studied in previous works, we have made our conclusions on oxide inclusion types based on the visual confirmation of EDX maps as well as results of previous studies of ODS steels. Eurofer ODS is the European steel for fusion program and TEM analysis of large oxides were carried out carefully [42-43]. 13.5 Cr ODS alloys were designed in Karlsruhe Institute of Technology to increase the corrosion resistance ODS steels for fusion [44-46]. ODS steeps on the base of FeCrAl matrix metal were widely designed in Japan [47-48].

This part is now addressed in text at line 158.

Figure 10: It is not clear what "cluster number" refer to?

Re: The “cluster number” refers to the number of each cluster found in the studied volume (ordered from largest to smallest). This has been clarified in text.

In table 3, the clusters composition are determined by APT?

This is correct, it was done through chemical analysis of the APT. This point is now clarified in the text.

Reviewer 2 Report

Comments and Suggestions for Authors

This work addressed a very important fundamental issue about ODS alloys - how to perform a reliable statistical analysis of two types of nano-inclusions (i.e. oxides and clusters), especially in a ferromagnetic matrix. By taking advantage of the magnetic properties of nano-clusters, the authors developed a SANS based technique to overcome this long-standing challenge. It turned out that the combination of TEM, APT and SANS analyses can best reproduce the experimental hardness values. Although the rationality of the conclusions fully relies on the so-called dispersed-barrier-hardening model, this work provided useful insights for improving the characterization of nano-particles in metallic materials.  The manuscript is generally well organized and well written. I would like to recommend  it to be accepted in present form.

Author Response

We highly appreciate the favorable opinion of Reviewer, and improve the
paper following suggestions of all Reviewers: 

The scale of figures 3 and 4 has been increased for better viewing.

A typo on line 276 was fixed.

On line 151 a better description of thickness evaluation of samples for TEM was given 

On line 156 a better explanation of oxide stoichiometry was given with new sources cited.

On figure 10 the term "cluster number" was better explained 

It was additionally mentioned that cluster compositions are indeed discovered by APT